# Laboratory Diagnosis of Porphyria

**DOI:** 10.3390/diagnostics11081343

**Published:** 2021-07-26

**Authors:** Elena Di Pierro, Michele De Canio, Rosa Mercadante, Maria Savino, Francesca Granata, Dario Tavazzi, Anna Maria Nicolli, Andrea Trevisan, Stefano Marchini, Silvia Fustinoni

**Affiliations:** 1Dipartimento di Medicina Interna, Fondazione IRCCS Cà Granda Ospedale Maggiore Policlinico, 20122 Milan, Italy; francesca.granata@policlinico.mi.it; 2Porphyria and Rare Diseases Centre, San Gallicano Dermatological Institute IRCCS, 00144 Rome, Italy; michele.decanio@ifo.gov.it; 3EPIGET-Epidemiology, Epigenetics, and Toxicology Lab, Department of Clinical Sciences and Community Health, University of Milan, 20122 Milan, Italy; rosa.mercadante@unimi.it (R.M.); dario.tavazzi@unimi.it (D.T.); silvia.fustinoni@unimi.it (S.F.); 4Servizio di Medicina Trasfusionale e Laboratorio Analisi, Laboratorio di Immunogenetica, IRCCS Ospedale “Casa Sollievo della Sofferenza”, 71013 San Giovanni Rotondo, Italy; mg.savino@operapadrepio.it; 5Dipartimento di Scienze Cardio-Toraco-Vascolari e Sanità Pubblica, Università Degli Studi di Padova, 35121 Padova, Italy; annamaria.nicolli@unipd.it (A.M.N.); andrea.trevisan@unipd.it (A.T.); 6Laboratorio Malattie Rare-Settore Porfirie, Dipartimento di Scienze Mediche, Chirurgiche, Materno-Infantili e Dell’Adulto, Azienda Ospedaliero-Universitaria Policlinico di Modena, 41125 Modena, Italy; stefano.marchini@unimore.it; 7Environmental and Industrial Toxicology Unit, Fondazione IRCCS Ca’ Granda Ospedale Maggiore Policlinico, 20122 Milan, Italy

**Keywords:** porphyria, ALA (5-aminolevulinic acid), PBG (porphobilinogen), porphyrins, HPLC (high-pressure liquid chromatography), MLPA (multiplex ligation-dependent probe amplification), NGS (next-generation sequencing), EPNET (European Porphyria Network), diagnosis

## Abstract

Porphyrias are a group of diseases that are clinically and genetically heterogeneous and originate mostly from inherited dysfunctions of specific enzymes involved in heme biosynthesis. Such dysfunctions result in the excessive production and excretion of the intermediates of the heme biosynthesis pathway in the blood, urine, or feces, and these intermediates are responsible for specific clinical presentations. Porphyrias continue to be underdiagnosed, although laboratory diagnosis based on the measurement of metabolites could be utilized to support clinical suspicion in all symptomatic patients. Moreover, the measurement of enzymatic activities along with a molecular analysis may confirm the diagnosis and are, therefore, crucial for identifying pre-symptomatic carriers. The present review provides an overview of the laboratory assays used most commonly for establishing the diagnosis of porphyria. This would assist the clinicians in prescribing appropriate diagnostic testing and interpreting the testing results.

## 1. Introduction

Porphyrias comprise a group of eight metabolic disorders originating from a genetically caused catalytic dysfunction of the enzymes involved in the heme biosynthesis pathway (Figure 1) [1]. Dominant or recessive inherited mutations in any of the genes encoding these enzymes lead to disturbance in heme synthesis along with the pathological accumulation and measurable excretion of the intermediates of the heme biosynthesis pathway [2].

An accumulation of the following two different kinds of metabolites may occur: one is the porphyrin precursors, such as 5-aminolevulinic acid (ALA) and porphobilinogen (PBG), which are linear non-fluorescent molecules, and the other kind is the porphyrins, such as uroporphyrins (URO), coproporphyrins (COPRO), and protoporphyrins (PROTO), which are circular molecules that emit fluorescence signals when excited. The accumulation of these metabolites occurs in different biological samples depending on their chemical properties. Since the hydrophobicity gradient increases as the heme synthesis progresses, the most hydrophilic metabolites (ALA, PBG, URO, COPRO) occur mainly in urine, while the relatively hydrophobic ones (COPRO, PROTO) occur in feces [3].

Porphyrins are the oxidized products of porphyrinogens, which are the actual substrates of the enzymes involved in heme biosynthesis. Porphyrins exist in different isomers depending on the arrangement of the substituents acetate (A), propionate (P), methyl (M), and vinyl (V) of the four pyrroles of the porphyrin ring. The most common isoforms are isoform III with asymmetrically arranged substituents and isoform I with symmetrically arranged substituents. The biosynthesis of heme involves the isoforms III since hydroxymethylbilane (HMB) is transformed, by the action of uroporphyrinogen III synthase (UROS), in uroporphyrinogen III. This undergoes subsequent decarboxylation by uroporphyrinogen decarboxylase (UROD) to form hepta-, hexa-, and penta-carboxyl porphyrinogen III, and finally, coproporphyrinogen III. However, in physiological conditions, a fraction of HMB escapes from the catalytic action of the UROS, and results in a non-enzymatic conversion of HMB to the uroporphyrinogen I isomer. Uroporphyrinogen I may subsequently undergo decarboxylation by the action of UROD to form hepta-, hexa-, and penta-carboxyl porphyrinogen I, and finally, coproporphyrinogen I; however, the reaction cannot proceed beyond this step to form heme as the next enzyme in the pathway—coproporphyrinogen oxidase (CPOX)—is stereospecific for the isomer III (Figure 2). Therefore, the isomers I accumulate in the tissue and are excreted as porphyrins I. A large presence of porphyrins I as well as an abnormal isomer I/isomer III ratio in biological fluids are relevant in defining the presence of porphyria [4].

Physiologically, heme synthesis is accomplished by the sequential action of eight enzymes and the whole process is finely regulated. The intermediate substrates other than the final product heme might exert a regulatory effect on the enzymes involved in the pathway [5]. Moreover, the catalytic capacity of the enzymes in different segments of the pathway varies strongly, leading to variations in substrate pressure. Therefore, the dysfunction of a specific enzymatic activity may cause an accumulation of the substrate immediately before the blockage and of other previous metabolites of the same proceeding line [6].

Specific combinations of metabolites (patterns) are associated with specific clinical features; sufficient evidence suggests that porphyrin precursors cause symptoms by injuring neurons, while porphyrins cause symptoms by injuring the skin and the liver [7]. Usually, the levels of porphyrin precursors increase during the presentation of acute neurovisceral attacks induced by exogenous factors, such as medications, nutritional status, stress, exogenous hormones, infection, etc. Moreover, depending on the patient, these levels may return to the normal ranges or remain persistently elevated during the clinically asymptomatic phase [8]. On the contrary, the accumulation of porphyrins is typically chronic, causing persistent cutaneous photosensitivity or skin fragility as a consequence of prolonged exposure to the sun [9].

The clinical diagnosis of porphyria is difficult as the condition may manifest with a broad and unspecific spectrum of clinical symptoms mimicking several other disorders. However, once suspected, the diagnosis of porphyria may be established using several laboratory tests available for symptomatic patients [10,11]. The laboratory diagnostic procedures comprise the following three important sequential steps: (i) the biochemical measurement of porphyrin precursors and porphyrins in the plasma, blood, urine, and feces, including both qualitative screening and quantitative confirmatory tests; (ii) the determination of specific enzymatic activities in erythrocytes or immortalized lymphocytes; and (iii) mutational analysis using both classical and next-generation molecular genetics techniques [12]. In the present review, an overview of the laboratory analysis methods used most commonly for establishing the diagnosis of porphyria is provided to assist the clinicians in prescribing appropriate diagnostic testing and interpreting the results.

## 2. Qualitative Screening Tests

### 2.1. Plasma Scan

Plasma scan is a simple, rapid, and inexpensive screening test that allows for the detection of the presence of increased amounts of URO, COPRO, and PROTO in the plasma, based on the capacity of these photoactive molecules to fluoresce when irradiated with wavelengths close to 400 nm. Once the plasma samples are obtained, the assay may be performed and completed within a few minutes. Briefly, 100 µL of plasma, diluted five-fold in a 0.25 M potassium phosphate-buffered saline (PBS) at pH 6.7 is scanned inside a classical fluorescence spectrophotometer at the excitation wavelength of 405 nm [13]. Otherwise, 1 mL of freshly drawn plasma may be centrifuged at 14000 rpm for 1 min and then 2 µL of the supernatant may be used directly for analysis in NanoDrop 3100, using a UV LED for excitation (maximum at 365 nm) [14]. In both methods of conducting the test, the fluorescence emission spectrum is recorded in the wavelength range of 580 to 700 nm.

The fluorimetric emission scanning of plasma is a qualitative test, in which the presence or absence of porphyrins is detected as a positive or negative result, respectively. If positive, the analyst then identifies the wavelength at which the maximum emission peak is obtained, as this wavelength is typical of specific porphyria conditions. The emission maximum obtained within 626–628 nm is specific to variegate porphyria (VP) [15,16], while a peak within 634–636 nm is characteristic of erythropoietic protoporphyria (EPP) and X-linked protoporphyria (XLP) [13]. A fluorescence emission maximum within 618–622 nm corresponds to ALA dehydrase (ALAD) deficiency porphyria (ADP), acute intermittent porphyria (AIP), hereditary coproporphyria (HCP), congenital erythropoietic porphyria (CEP, Gunther disease), porphyria cutanea tarda (PCT), and hepatoerythropoietic porphyria (HEP) (Table 1) [17].

The intensity of the fluorescence peak varies with the activity status of the disease and is always positive in all symptomatic patients, including those with a suspected attack of acute porphyria in progress (ADP, AIP, VP, HCP) [18] and those with chronic high excretion of porphyrins (CEP, PCT, HEP, EPP). However, only 50% of asymptomatic VP carriers are detected using this method, which renders the plasma scan test inefficient for family studies involving the detection of gene carriers in both adults and children [19,20]. Moreover, porphyrins in the plasma sample of patients with EPP and VP are particularly labile if exposed to light. Therefore, without adequate light protection, the levels of measurable porphyrin decline rapidly, resulting in false negatives [21]. In addition, PCT patients treated with phlebotomy and patients with acute porphyria in the latent phase may not present any peak due to normal porphyrin concentration in their plasma samples. The absence of any peak in the emission spectrum, in a well-preserved sample, would imply that the diagnosis of active porphyria may be rejected.

On the contrary, patients undergoing maintenance hemodialysis therapy [22,23] or those co-infected with immunodeficiency virus (HIV) and hepatitis C virus (HCV) [24] may develop a secondary elevation of plasma porphyrin along with cutaneous lesions, similar to those observed in genetic PCT patients, and as a consequence, a characteristic fluorescence peak appears at 618–620 nm. Therefore, this test might be useful for monitoring the activity of all the conditions referred to as pseudoporphyria, which do not result from an enzymatic absence of genetic origin.

In conclusion, routine plasma fluorometric screening could be performed as the first diagnostic approach in cases with high levels of porphyrins in the plasma, such as in those with acute attack and the ones with chronic cutaneous porphyria. However, considering that several porphyria conditions have the same characteristic emission peak, the positive results should be confirmed with specific quantitative analysis of porphyrins to establish the correct diagnosis.

### 2.2. Fluorocytes

The excess of porphyrins in erythrocytes may be detected by the presence of fluorescent red cells (fluorocytes), which may be identified using three qualitative screening tests. The first one is to record, similar to the approach in the plasma scan test, the fluorescence emission spectrum of the erythrocytes diluted in buffered 0.9% saline at a ratio of 1:2000, in the wavelength range of 550–650 nm using 400–415 nm as the excitation wavelength range throughout, in a fluorescence spectrophotometer. The emission peaks for URO, zinc–protoporphyrin (ZnPP), and metal-free protoporphyrin (PPIX) are obtained at 620, 587, and 630 nm, respectively [21,22,23,24,25].

Otherwise, a drop of freshly obtained blood could be smeared on a glass slide and viewed using a fluorescence microscope under ultraviolet light (380–450 nm) [26]. Although an automated imaging system, which improved the traditional fresh blood film method, was described to facilitate population screening in EPP, this method requires special equipment and expertise and, therefore, is limited to specialized centers [27]. Moreover, fluorescence microscopy presents problems associated with quantification, and the radiation focused on individual erythrocytes rapidly photodegrades protoporphyrin, thereby causing evanescence of the fluorescence [28].

On the contrary, cytofluorimetric analysis is a rapid and sophisticated method for the measurement of the percentage of fluorocytes in the samples. In this method, a sample containing 10 µL of ethylenediaminetetraacetic acid (EDTA)-whole blood diluted in 10 µL of phosphate-buffered solution at pH 7.2 is processed directly in a flow cytometer without staining. Fluorescent erythrocytes emit red fluorescence at λ > 620 nm when excited at 488 nm using an argon laser. The results are expressed as the percentage of cells beyond the threshold of the autofluorescence of the control subjects [29]. In contrast to fluorescence microscopy, this method does not present the problem of subjectiveness in the evaluation of the intensity of fluorescence in a possibly small fraction of rapidly fading fluorocytes.

Although high levels of fluorocytes may be detected in various forms of porphyria using this method, the test should be predominantly used in EPP [30], in which the symptoms are not as severe and evident as in CEP and HEP. However, the test only quantifies the percentage of fluorescent erythrocytes and not the actual amount of total protoporphyrins. Even though the results of cytofluorometry are reported to correlate well with those of the quantitative HPLC test [31], the absolute quantification of porphyrins is necessary to establish the diagnosis of EPP. Nonetheless, once the diagnosis is established, cytofluorometry might prove to be useful in the screening and follow-up of patients.

### 2.3. Hoesch Test

Rapid tests for detecting the presence of increased PBG in the urine are of great significance in the early diagnosis of an acute attack of porphyria [32]. Initially, the test used in this setting was the one developed by Watson and Schwartz in 1941, which was subsequently modified in 1964 [33,34]. Although this test was termed as a screening test, it is actually a multistep procedure, and the number of reagents required, the time consumed for performing the test, and the expertise required for appropriately conducting and interpreting the test results have limited its use in clinical wards. Therefore, in current times, this test has been replaced by a simpler and rapid test, which is less prone to misinterpretation by inexperienced personnel, known as the Hoesch test [35].

In the Hoesch test, two drops of fresh urine are added to 1 mL of Ehrlich’s reagent (solution of 0.2 g of p-dimethylaminobenzaldehyde (DMAB) in 10 mL of 6M HCl). In the case of increased PBG in the urine sample, as the urine contacts the reagent, a cherry-red color develops immediately at the surface of the solution and then throughout the tube under agitation [36]. The sensitivity of the test is typically 10 mg/L, while the upper normal limit for urinary PBG is approximately 2.5 mg/L. Therefore, small increases in PBG might be missed when using this test, although these are not of practical importance when the test is being performed for symptomatic patients suspected of having an acute attack, as these patients present markedly increased levels of urinary PBG in the range of 10 to 20 times higher than the upper normal value [37]. 

Moreover, the Hoesch test procedure appears to have higher specificity compared to the Watson–Schwartz test, as the former does not involve an interfering reaction in the presence of urobilinogen in patients with liver disease [38]. The other possible sources of false-positive Ehrlich reactions have not been investigated, and the potential inhibitors of this reaction have also not been evaluated thus far [39,40]. Using the Hoesch test, it is possible to detect the presence of high levels of PBG in the urine accurately and rapidly [41]. Moreover, it is reported that Ehrlich’s reagent may be stored in a clear glass container for a minimum of nine months without a loss of activity [38]. These data confirm the efficacy and feasibility of this first-line qualitative test in detecting an acute attack in progress. Nonetheless, further investigative procedures to confirm the diagnosis and identify the type of porphyria should be performed.

## 3. Quantitative Confirmatory Tests

### 3.1. ALA and PBG Determination

The quantification of ALA and PBG forms the first line of laboratory testing for acute porphyria in the event of potentially life-threatening acute neurovisceral attacks [42]. For decades, the collection of urine samples over a period of 24 h was the rule leading to a harmful delay of the necessary therapeutic measures. Meanwhile, this procedure has been replaced by the spot urine sample procedure that is considered to be sufficient to estimate the activity of acute porphyrias and allow for a decision on therapeutic intervention. To date, in specialized porphyria diagnostic laboratories, ALA and PBG are commonly quantified after purification, from a spot urine sample, using two commercially available anion-exchange and cation-exchange columns (ClinEasy^®^ Complete Kit for ALA/PBG in Urine, Recipe GmbH, Munich, Germany; ALA/PBG by Column Test, Bio-Rad Laboratories, Hercules, CA, USA). This approach enables the selective purification of ALA and PBG, which removes the sample matrix, thereby preventing possible interferences from other compounds [43]. In several certified European laboratories (EPNET), the preferred choice is the ClinEasy^®^ Complete Kit for ALA/PBG in Urine of Recipe for the quantification of both ALA and PBG in the urine. Briefly, urine samples are passed through two overlapping columns—the top column, containing an anion exchange resin, adsorbs PBG, and the ALA passes through this top column and is subsequently retained by the cation exchange column at the bottom. The adsorbed PBG is eluted from the top column using acetic acid and mixed with Ehrlich’s reagent, upon which the solution develops a purple color, whose intensity is proportional to the amount of the metabolite in the solution. The retained ALA is eluted using sodium acetate, and after derivatization with acetyl-acetone at 100 °C, it is converted into a monopyrrole, which is the measurable form capable of reacting with Ehrlich’s reagent. The absorbance of both PBG and ALA is measured against the blank reagent at 533 nm using a UV–VIS spectrophotometer. Metabolite concentrations are calculated through comparison to the appropriate calibrators (Urine Calibrator Lyophil RECIPE GmbH, Munich, Germany). The results are validated using normal and pathological controls (ClinChek Urine Control L1, L2, RECIPE GmbH, Munich, Germany) reconstituted in high purity water and stored in single-dose aliquots at −20 °C until used. The concentration values are expressed as µmol/mmol creatinine. As demonstrated by Stauch Th and colleagues in 2017, in the occasion of the international congress of porphyrins and porphyria in Bordeaux, ALA and PBG are considered over the normal limits if the concentration values are >5 and >1.5 µmol/mmol creatinine, respectively. A high level of ALA and a normal level of PBG indicate either the rare ALAD deficiency porphyria or the more common heavy-metal intoxication and hereditary tyrosinemia type I caused by the inhibition of ALAD by lead and succinyl-acetone, respectively [44,45]. High levels, of up to 20–50 times the normal values, of both the metabolites establish the diagnosis of an acute attack, while moderate ALA and PBG levels could indicate VP and HCP [6]. The majority of the patients affected by acute porphyria after the symptomatic period may revert completely, while a few might become clinically asymptomatic along with persistent moderate increments in the heme precursors.

Over the last few decades, the technique of liquid chromatography-tandem mass spectrometry (LC-MS/MS) has been employed in numerous clinical biochemistry applications. In comparison to traditional methods, LC-MS/MS has major advantages of higher analytical sensitivity, specificity, and diagnostic reliability. About porphyria, the LC-MS/MS technique has been applied successfully for the simultaneous quantification of ALA and PBG in urine and plasma samples [46,47,48,49].

Notably, significantly lower levels of urinary ALA and PBG could be measured using MS-based methods, particularly for healthy individuals, confirming that these methods have a higher selectivity compared to the colorimetric ones [43]. The difference is probably due to the presence of contaminant molecules in the urine, which react with Ehrlich’s reagent. Recently, LC-MS/MS measurements of a large number of healthy subjects established the upper limit of the normal values (ULN) of ALA and PBG as 1.47 and 0.137 µmol/mmol creatinine, respectively [50], and it was reported that during an acute attack, these values could reach 40 and 55 µmol/mmol creatinine, respectively [48]. Moreover, sample preparation using a solid-phase extraction (SPE) system allows the detection of concentrations as low as 0.05 µM [49]. Therefore, measurements of porphyrin precursors in plasma and tissue samples have become achievable. Floderus et al. quantified the plasma levels of ALA and PBG in 10 asymptomatic AIP patients and reported the mean concentrations to be 1.7 and 3.1 µmol/L, respectively, which were significantly higher than those of the healthy subjects (0.38 and <0.12 µmol/L, respectively) [46]. However, during an acute attack of porphyria, ALA and PBG concentrations may increase dramatically, up to 13 µmol/L [47].

Since the variation over the time of the plasma concentrations of ALA and PBG is reported to be highly correlated with urinary concentrations [46], these measurements may contribute to the monitoring of AIP patients during the course of an acute attack [51], particularly if the patients are in an anuric state. Plasma ALA and PBG levels could also be useful in evaluating the safety and the pharmacokinetic effects of the existing or future therapies in AHP patients [52]. However, not all hospitals are equipped with LC-MS/MS, restricting this technique to a few specialized centers.

### 3.2. Measurement of Urine Porphyrins

The differential diagnosis of porphyria relies on the measurements of porphyrins and relative isomers in urine, feces, and plasma. As plasma porphyrin separation finds a practical application only in patients with renal failure, this topic is not treated in this review.

Although various methods have been developed for the analysis of urine porphyrins, reverse-phase high-pressure liquid chromatography coupled with fluorescence detection (FLD-HPLC) has become the gold standard method for this purpose [53,54,55]. The success of this technique is attributed to its capacity to separate the physiologically relevant porphyrins as free acids and resolve type I and type III porphyrin isomers simultaneously [56]. Since the fluorescence detection enhances the specificity of the analytic method, matrix extraction procedures are not required, and acidic urine samples could be directly injected.

Typical chromatographic runs are performed on a C18-bonded silica stationary phase using a linear gradient elution system from 10% (v/v) acetonitrile in 1M ammonium acetate, pH 5.16 (phase A) and 10% acetonitrile in methanol (phase B). Porphyrins containing from two to eight carboxylic groups, including the resolution of type I and type III isomers could be achieved in less than 30 min. The excitation and emission wavelength ranges used are usually around 395–420 and 580–620 nm, respectively. Direct standardization is obtained by comparison to chromatographic runs of a suitable calibration standard that are now commercially available (RECIPE GmbH, Munich, Germany; Chromsystems GmbH, Gräfelfing, Germany). The total urine porphyrin measurement is calculated as the sum of the chromatographic fractions and the single fractions as relative percentages. As spot urine samples are commonly used, the results should be normalized on the creatinine concentration.

In normal subjects, the total porphyrins excretion is found below 35 nmol/mmol creatinine, COPRO predominate on URO, while hepta-, hexa-, and penta-carboxyl porphyrins are present only in small amounts. Moreover, relative concentrations of COPRO isomer III are higher than COPRO isomer I.

In porphyric patients, the excretion of urinary porphyrins varies in relation to the enzymatic defect underlying the particular type of porphyria and with respect to the disease stage [57]. URO, both type I and type III isomers, and hepta- type III are evidently raised in PCT and HEP, since the type I isomers of URO and COPRO are detected in CEP almost exclusively. In acute hepatic porphyrias (AIP, VP, and HCP), porphyrins’ excretion is extremely variable from normal values in the asymptomatic phase to very high values observed during the acute exacerbation of the disease. In AIP, a pattern of the marked elevation of URO I and III isomers, along with less pronounced elevations of COPRO III and I, is detected, while HCP and VP demonstrate a marked elevation of COPRO III. In EPP, total urine porphyrins are normal; however, abnormal chromatographic profiles are frequently observed. In particular, higher relative amounts of COPRO I are commonly observed as a consequence of potential hepatic implications. In fact, increased levels of porphyrins excretion may also occur in several physio-pathological conditions including hereditary hyperbilirubinemias, toxic syndromes or liver diseases [58]. Examples of porphyrins’ pattern of normal and pathological subjects are provided in Figure 3.

Recently, protocols using mass spectrometry (LC-MS/MS) to separate and detect porphyrins have also been reported to facilitate the clinical diagnosis of porphyria [48]. However, the low prevalence of this equipment in hospitals restricts the application of such protocols to the identification and characterization of unknown porphyrins in research.

### 3.3. Analysis of Fecal Porphyrins

Two different systems may be employed for the analysis of fecal porphyrins. First, the total fecal porphyrins can be quantified using a spectrophotometric [58,59] or fluorimetric [60] method and then separated by HPLC to obtain porphyrin patterns [61]. Second, total fecal porphyrins can be calculated as the sum of fractions following HPLC analysis [62]. The former approach is considered more suitable for routine use in clinical analysis, while the latter is technically more correct and, therefore, allows for quantification with a higher reliability [63].

In the widely employed method reported by Lockwood et al., porphyrins are extracted from a small sample of feces in the aqueous acid phase using the solvent partition [59]. Briefly, 25–50 mg of feces is processed by the sequential addition of 1 mL of concentrated HCl to dissolve the organic matrix, 3 mL of diethyl ether to eliminate the contaminants—chlorophylls and carotenoid pigments, and 3 mL of water to avoid any alteration in protoporphyrin. The hydrochloric acid extract is then analyzed by performing a spectrophotometric scan between 370 and 430 nm, which includes the Soret region. After background signal subtraction, the absorbance measured at the maximum peak is used for calculating the total porphyrin content. It is also necessary to separately evaluate the water percentage in the feces sample to report the result as nmol/g dry weight (total fecal porphyrin normal value < 200 nmol/g dry weight) [59]. Although no commercial reference material is available, an internal quality control (ICQ) could be prepared from the patient’s specimens to check the day-to-day reproducibility. The obtained hydrochloric acid extracts may be directly injected into HPLC systems for subsequent characterization using the same protocol as the one used for urine porphyrins chromatography. The calibration solution is prepared by adding appropriate concentrations of meso- and proto-porphyrin to the urine calibrators.

The fecal excretion of porphyrins increases in hepatic PCT, HCP, and VP, in erythropoietic CEP and HEP, and sometimes in EPP and XLP. The detection of the specific patterns of fecal porphyrins allows the differentiation of these enzymatic disorders that otherwise share similar clinical presentation and overlapping biochemical characteristics. Feces samples from healthy subjects, as well as from porphyria patients, contain varying amounts of dicarboxylic porphyrins, particularly deutero-, pempto-, and meso-porphyrins, in addition to protoporphyrin. These dicarboxylic porphyrins lack a diagnostic relevance, depending on the gut microflora [64] and even on the diet [65]. Moreover, gastrointestinal bleeding may interfere with feces analysis, causing an anomalous increase in the levels of protoporphyrin and the dicarboxylic porphyrins derived from it [66]. Tricarboxylic porphyrins remain generally undetected, although the presence of harderoporphyrin in feces [67] played a major role in the identification of a variant of homozygous HCP (harderoporphyria). Uroporphyrin is excreted prevalently in the urine and may be detected in feces only in trace amounts.

Fecal porphyrins profiles of PCT and HEP are the most complex, with a wide range of peaks. It is possible to recognize the COPRO I isomer, which always prevails over COPRO III, although increasing signals corresponding to epta-, esa-, and penta-porphyrins, with the prevalence of isomer III, are also observed. Since UROD is the enzyme that catalyzes the sequential decarboxylation of uroporphyrin to coproporphyrin, its activity deficit leads to the accumulation of all the porphyrin intermediates in PCT. In HEP, the activity of UROD is severely compromised and the porphyrins containing a higher number of carboxylic groups are represented more. PCT chromatographic profiles are also characterized by the presence of isocoproporphyrin and its derived metabolites hydroxy-, keto-, deethyl-, and dehydro-isocoproporphyrin. These molecules are commonly identified as a diagnostic sign of symptomatic PCT [64]; nevertheless, diagnoses based on the presence of isocoproporphyrin in the feces are prone to be erroneous.

The chromatographic profiles of the fecal porphyrins of AIP and ADP patients are similar to those of healthy subjects. On the other hand, the fecal porphyrin chromatograms of HCP patients are easily recognizable by the presence of a prominent peak corresponding to COPRO III. The peaks for both PROTO and COPRO III are elevated in the chromatographic profiles of VP patients. In particular conditions, such as hepatitis or drug consumption, which inhibit the UROD activity, HCP and VP patients may exhibit fecal porphyrin patterns quite similar to those of PCT patients, including the isocoproporphyrin series. Since the relative abundance of COPRO III is always higher compared to isomer I in VP and HCP, while the opposite is true for PCT, the ratio between the coproporphyrin isomers serves as a suitable diagnostic parameter [68,69]. The fecal porphyrins pattern in CEP contains an elevated peak corresponding to COPRO I. Finally, the chromatograms of EPP and XLP patients present an elevated PROTO peak. Examples of the porphyrins pattern of normal and pathological subjects are provided in Figure 4.

### 3.4. Erythrocyte Porphyrins Measurement

Initially, to assay the total erythrocyte porphyrins, fluorometric methods involving the acid extraction procedure that converted ZnPP to its metal-free form PPIX were used [70]. Later, when it became possible to detect ZnPP in the whole blood samples without prior acid extraction, hematofluorometry methods were developed for this assessment [71,72]. Finally, the application of high-performance liquid chromatography (HPLC) coupled with fluorometric detection to separate and quantify unchelated PPIX and ZnPP simultaneously commenced [73,74]. A method using derivative variable-angle synchronous fluorescence (DVASF) for determining PPIX and ZnPP simultaneously in a whole blood sample, avoiding the spectral compensation factor for PPIX and the chromatographic separation, has been also reported [75]. A simple, rapid, and specific HPLC method is described here. Whole blood samples are collected in vacutainer tubes containing the anticoagulant K3EDTA and then stored at −20 °C in the dark. At the time of analysis, samples are thawed to room temperature and mixed well, followed by the dilution of 60-microliter aliquots in 200 µL of lysing solution (4% aqueous formic acid) and the extraction of porphyrins using 900 µL of acetone. An aliquot of the extracted porphyrins is injected directly into the HPLC system, using a Chromsystem C-18 column (Chromsystems GmbH, Gräfelfing, Germany) for chromatographic separations. ZnPP and PPIX are eluted in isocratic conditions (90% methanol in an aqueous 1% acetic acid solution) at 42 °C and subsequently detected using fluorescence excitation/emission wavelengths of 400/620 nm (1–7 min) and 387/633 nm (7–15 min), respectively. In order to quantify PPIX and ZnPP, homemade calibration curves are used, and the analytes are observed to be linear in the concentration ranges of 1.5–50 and 2–100 µg/dL, respectively. The results are reported as the sum of the PPIX and ZnPP peaks, and the concentrations in the blood are expressed as both µg/g Hb and relative percentage of each porphyrin. The normal level for the sum is <3 µg/g Hb and the normal percentage ranges are ZnPP 80–90% and PPIX 10–20%. Erythrocyte porphyrins are abnormally increased in EPP, XLP, CEP, and HEP, although the percentage of each porphyrin differs among these disorders, as described in Table 2. It is noteworthy that elevated erythrocyte protopophyrins along with normal ZnPP support the diagnosis of the classical form of EPP, while an elevation in both components with balanced percentages occurs in the XLP. High levels of erythrocyte porphyrins also occur in the case of exposure to lead from both environmental and occupational conditions [76], in iron deficiency [77], as well as in sideroblastic anemia, all of which are conditions associated with elevated Zn PP. In CEP and HEP, the predominant porphyrins may be uroporphyrins or ZnPP, depending on the phenotype expression.

## 4. Enzymatic Assays

Enzyme deficiencies leading to porphyrias are detectable in tissue samples, such as comprising the liver tissue, white blood cells, lymphoblasts, and cultured fibroblasts, although this detection requires the use of invasive techniques to obtain sufficient amounts of samples from the patients [78]. In this context, enzymatic assays, conducted using tissue samples, play only a minor role in the routine diagnosis and management of porphyrias and remain limited to research procedures [79]. This is the case of mitochondrial enzymes, such as ALAS1, CPOX, PPOX, and FECH, which require Epstein–Barr virus-immortalized lymphoblastoid cell lines to dose their activities or HPLC assays to increase the sensitivity [80,81,82,83,84]. On the other hand, the quantification of porphyrins in body fluids is feasible and the preferred choice for the differential diagnosis of porphyrias. Four cytosolic enzymes, namely, ALAD, PBGD, UROD, and UROS, are easy to detect in the erythrocytes and, subsequently, recoverable in quantities sufficient for dosing. However, in the diagnostic procedure, only the first two of these four enzymes are commonly used. Although UROD activity is useful for differentiating the sporadic and familial cases of PCT, its results are technically demanding [85]. Therefore, a diagnostic model that includes the clinical and biochemical data of patients, such as sex, age at the time of diagnosis, other family members having overt PCT, uro-/hepta-porphyrin ratio, ferritin-glutamyltransferase, alanine aminotransferase, alcohol consumption, and hepatic viral infections, is preferred in clinical practice [86,87]. Moreover, due to the rarity of CEP patients worldwide, the possibility of diagnosing CEP using other biochemical assays, and the severity of the symptoms of CEP, the UROS enzymatic activity-based assay is of little utility in the diagnosis of porphyria.

### 4.1. ALAD Enzyme Activity

Delta-aminolevulinic acid-dehydrase (ALAD) is the second enzyme of the heme biosynthesis pathway, which catalyzes the biosynthesis of PBG via the condensation of two molecules of ALA. An ALAD-based assay is conducted on erythrocytes and involves the reaction of its enzymatic product PBG with Ehrlich’s reagent to obtain a derivative compound that is detectable calorimetrically [88,89]. A coupled enzyme assay has also been described in which hydroxymethylbilane (HMB) is produced by the action of the downstream enzyme porphobilinogen deaminase (PBGD) using PBG, the product of the ALAD reaction, as the substrate [90]. HMB spontaneously cyclizes to uroporphyrinogen I and is detectable in spectrofluorometry as oxidized uroporphyrin I. Recently, tandem mass spectrometry was applied to the ALAD assay [91]. The incubation of red blood cells (RBC) lysate with ALA was followed by the quantitative in situ conversion of PBG into its butyramide and the extraction of the derivative product into a mass spectrometer-friendly liquid solvent. The extracted product was quantified by employing electrospray ionization tandem mass spectrometry, using a deuterium-labeled internal standard.

The quantification of the ALAD activity must performed within 24 h from the withdrawal of blood, as ALAD is quite unstable at 4 °C and deteriorates rapidly even at −20 °C [79]. For the determination of ALAD activity, 200 μL of Na-heparin anticoagulated blood is diluted to 1.5 mL using dH2O to obtain a crude (1:7.5 v/v) lysate, which is then incubated at 37 °C for 5 min. One mL of 50 mM aminolevulinic acid in 0.1 M sodium phosphate buffer (pH 6.4) is added to this lysate and the reaction mix is incubated at 37 °C for 1 h. A blank control tube is prepared, which contains 1 mL of 0.1 M sodium phosphate buffer alone. The reaction is terminated by adding 1 mL of 20% trichloroacetic acid containing 2.5% HgCl_2_, followed by centrifugation at 2500 rpm for 15 min. One volume of Erlich’s reagent is added to each of the supernatants obtained after the centrifugation of blank and sample reaction mixtures, and then absorbance at 555 nm is measured in a UV–VIS spectrophotometer. The PBG, produced at the end of the ALAD reaction, is quantified using an extinction coefficient of 62 cm^2^/mmol. The enzyme activity is expressed as nmol PBG/h/mL RBC. The normal level of the ALAD activity is >20 nmol PBG/h/mL RBC.

The inhibition of the ALAD activity and variation in the concentrations of certain heme biosynthesis intermediates (ALA in the urine, blood, or plasma; COPRO the urine; ZnPP in the blood) are also biomarkers of the early effects of toxic heavy metals, such as lead and mercury, in the bone marrow and the nervous system [92,93]. Furthermore, a severe ALAD deficiency (<10 nmol PBG/h/mL RBC) caused by homozygous or compound heterozygous genetic defects in the *ALAD* gene is responsible for the extremely rare ALAD deficiency porphyria (ADP) [94].

### 4.2. PBGD Enzyme Activity

Porphobilinogen deaminase (PBGD) is the third enzyme in the heme biosynthesis pathway and catalyzes the biosynthesis of the linear pyrrole HMB by the condensation of four molecules of the pyrrole PBG. Subsequently, HMB is acted upon by UROS and converted to uroporphyrinogen III, which is easily detectable. Erythrocyte PBGD is usually measured through spectrofluorometric methods using PBG as the substrate [78,95,96]. Recently, an assay based on tandem mass spectrometry was reported for this purpose [97]. Thus far, there is no international standardization established for the assessment of PBGD activity. However, Puy et al. (1997) described a practical and reliable method to measure the activity of the PBGD enzyme [95]. In this method, RBCs obtained from EDTA-anticoagulated blood are lysed by adding 10 volumes of 0.1 M of TRIS-HCl buffer (pH 8.0) containing 0.2% Triton X-100, followed by the measurement of the Hb concentration using the standard cyanmethemoglobin method. An aliquot of 25 µL of the RBC lysate is mixed with 200 µL of Tris-HCL buffer and 25 µL of 1 mM PBG (substrate of the reaction). After incubation in the dark for 60 min at 37 °C, the reaction is terminated by adding 1 mL of 10% trichloroacetic acid. The mix is then centrifuged at 10,000× *g* for 5 min, and the fluorescence emission of the resultant supernatant is measured at the excitation and emission wavelengths of 405 and 655, respectively. Enzyme activity is expressed as pmol URO/h/g Hb. PBGD activity in healthy subjects ranges between 75 and 170 pmol URO/h/g Hb.

The application of this assay, which is available in several specialized laboratories, would contribute to the identification of patients suspected with AIP and of asymptomatic AIP carriers in family studies. In the commonly occurring Type I AIP, up to 50% of enzyme deficiency is observed in erythroid and hepatic tissues. On the other hand, in AIP Type II, which is a sub-variant of AIP that occurs in approximately 5% of all AIP cases, a diminished enzyme activity is detectable only in the non-erythroid tissues, while the enzyme activity in erythrocytes is normal [98,99]. Slight reductions in activity could also be observed in HCP and VP patients, likely due to the negative allosteric effects exerted by proto- and coproporphyrinogen on PBGD [100]. Recently, an improved PBGD activity assay has been also reported [101].

## 5. Genetic Testing

### 5.1. DNA Sequence Analysis

DNA analysis is considered the “gold standard” method for the diagnosis of genetic disorders. Various molecular approaches, such as Restriction Fragment Length Polymorphism (RFLP), Single Strand Conformation Polymorphism (SSCP), Denaturing Gradient Gel Electrophoresis (DGGE), and Denaturing High-Performance Liquid Chromatography (DHPLC), have been employed in previous years to assist the diagnosis of different forms of porphyria. However, owing to the remarkable molecular genetic heterogeneity, Sanger direct sequencing is the preferred method nowadays [102].

Although no germline mutations in the *ALAS1* gene are known to cause porphyria, ten different genes have been associated with the eight forms of porphyria [103]. Therefore, prior to requesting the DNA analysis, it is recommended to perform biochemical testing and provide all the data available regarding the major clinical characteristics and the primary sites of accumulation of porphyrins to define which of the genes to analyze. Moreover, multiple modes of inheritance are reported for these disorders, which renders it important to establish the number of mutations one expects to detect.

Three of the different forms of porphyria (AIP, HCP, and VP) are autosomal dominant disorders with a low clinical penetrance that is estimated to be approximately 1% of all mutation carriers [104,105]. However, extremely rare and severe cases of homozygous dominant forms of AIP [106], HCP [107], and VP [108] are also reported. The other three forms of porphyria (ADP, CEP, and EPP) are autosomal recessive conditions [109], although X-linked inheritance is also reported in the cases carrying *GATA1* and *ALAS2* mutations, which are responsible for CEP [110,111] and XLP [112,113], respectively. The remaining form of porphyria, known as familial porphyria cutanea tarda (Type 2), which is clinically indistinguishable from its sporadic Type 1 subtype, is inherited either as an autosomal dominant (fPCT) or as autosomal recessive (HEP) trait [114] (Table 3).

In order to identify the genetic defect causing porphyria, different protocols based on the use of different specific primer pairs targeting single exons containing relative splicing junctions and the 5′ end of the genes encoding the enzymes of the heme biosynthesis pathway have been developed and used worldwide in Sanger sequencing [115,116,117]. In brief, genomic DNA is extracted from peripheral blood leucocytes or epithelial cells using a buccal swab, by following the standard manual protocols involving saline solutions or automatic systems involving magnetic beads. The amount and the quality of DNA are evaluated spectrophotometrically and spectrofluorimetrically, respectively, and the regions of interest are amplified using the polymerase chain reaction (PCR). The amplified products are purified using any of the available systems based on filtration and, subsequently, sequenced using the BigDye Terminator sequencing kit. The obtained fragments are purified and then resolved on an Automatic Genetic Analyzer (Thermo Fisher Scientific, Waltham, MA, USA). The sequences are analyzed using specific software. The sequence variants, including small insertion and deletion, are detected using standard alignment-based variant calling methods against reference sequences. Finally, each of the identified variants is sequenced again independently.

A large number of disease-specific pathogenic variants, including both the loss-of-function (LOF) and the gain-of-function (GOF) mutations, have been identified in porphyria. Most of these variants are restricted to one or a few families, [118,119], while a few others have become widely distributed within a discrete population via founder effects [120,121,122,123,124]. A regularly updated list of identified mutations is available in the Human Gene Mutation Database (HGMD). However, not all the variants reported as disease-causing variants are supported by sufficient evidence in favor of their pathogenicity, and no public database documents the probable pathogenicity of these newly identified variants in the heme biosynthesis genes.

Therefore, an international collaborative project involving a team of porphyria diagnostic experts is establishing an online database that would collate biochemical and clinical evidence validating the pathogenicity of the already published and the newly identified variants causing different forms of porphyria [125]. Elevated levels of porphyrins and porphyrin precursors in the plasma, erythrocytes, urine, or feces are critical in determining the role of a novel mutation in causing the disease. The pathogenicity of a novel mutation may also be supported by in vivo reduced enzyme activity, in vitro expression studies, and the frequency of the mutation in the exome and genome databases, which would indicate that the mutation is common and, therefore, benign.

In conclusion, the genetic testing approach is the most accurate and reliable among the methods available for the diagnostic confirmation of a specific form of porphyria in symptomatic patients. Once a specific gene mutation is detected, other family members may also be tested to identify the at-risk asymptomatic carriers who require counseling to avoid symptoms or minimize disease complications. Although most mutations are identified in the coding regions, canonical splicing sites, or promoter regions of the genes causing porphyria, there is significant molecular genetic heterogeneity, due to which Sanger sequencing may not always be conclusive.

### 5.2. Multiplex Ligation-Dependent Probe Amplification (MLPA)

Intragenic deletions of the size of a few kilobases, which affect a single or several exons of the heme genes, may be missed when using routine DNA sequence analysis. Only a quantitative DNA analysis would be able to determine whether the PCR products are derived from a single copy of the gene or the normal two. Gene dosage analysis was first reported in regard to porphyria for identifying the deletions in the *FECH* gene [126,127]. This approach was designed to simultaneously amplify several exons of the target gene and two exons of the control genes using multiplex PCR and different fluorescent label primer pairs. The advent and the spread of the multiplex ligation-dependent probe amplification (MLPA) technique enabled the amplification and analysis of up to four genes simultaneously [128]. 

This method is based on the hybridization and ligation of two adjacently annealing oligonucleotides to form a probe specific for each target region to be analyzed. Each oligonucleotide pair (probe) is designed to contain common end sequences, which implies that all probes could be simultaneously amplified using one universal primer pair, thereby overcoming the low efficiency issue of standard multiplex PCR. Chemically synthesized MLPA probe sets, designed according to the recommendations of Stern et al. [129] and tested for each heme gene and their flanking regions, are reported for the quantification of single genes separately [130]. Thus far, two MLPA kits (P411 and P412), which use probes generated by cloning into specially developed M13 vectors, are available commercially for quantifying three (*ALAD, HMBS, PPOX*) and four (*CPOX, UROS, UROD, FECH*) genes simultaneously. Regardless of the type of probe used, the MLPA reactions are performed using an initial amount of 100 ng of genomic DNA and the reagents and recommendations of the EK1 MLPA reagent kit (MRC-Holland, Amsterdam, The Netherlands), being careful which universal primers to use.

The PCR products are separated easily based on their size by performing capillary electrophoresis on a Genetic Analyzer (Applied Biosystems, Warrington, UK). The trace data are analyzed and then quantified using Gene Mapper (Applied Biosystems) or Coffalyser (MRC-Holland) software. In both cases, after intra-sample and inter-sample normalization, a dosage quotient (DQ) value is obtained.

A locus with a double copy would result in a theoretical value of 1.0 (v.n. 0.80–1.20), while a locus with a deletion would result in a value of 0.5 (0.4–0.65). Deletions represent an important cause of human diseases and account for 5.6% of all mutations reported in the human gene mutation database (www.hgmd.cf.ac.uk, accessed on 1 July 2021). Approximately twenty large deletions (size > 0.5 kb) are reported in the heme genes, with the highest frequency in the *HMBS* [131,132,133,134] and *FECH* genes [127,135,136,137,138], followed by *UROD* [139,140], *CPOX* [141], *PPOX* [142], and *UROS* [143]. These data confirm that the heme genes are prone to such rearrangements and that MLPA is an optimal tool for detecting a single exon or long sequence deletion, thereby complementing the DNA sequence analysis.

### 5.3. Next-Generation Sequencing (NGS)

While DNA sequencing and MLPA of the genes in the heme biosynthesis pathway allows for the identification of the disease-causing mutations in almost all porphyria patients, the nature of the primary genetic defects in a few patients with clinical and biochemical symptoms might remain undetected. The reported sensitivities for the mutational analysis in different forms of porphyria are as follows: AIP, 98.1% (95.6–99.2%); HCP, 96.9% (84.3–99.5%); VP, 100% (95.7–100%); EPP 93.9% (89.4–96.6%); fPCT > 95%; and CEP, approximately 75% [144]. The mutations that are present deep within the introns of the heme genes, or those underlying the primer sequences or in other genes that are unknown might explain the loss of sensitivity in certain types of porphyria.

The roles of *ALAS2*, *GATA1*, and *CLPX* genes in the pathogenesis of porphyria have been identified [145,146]. This strongly suggests that future studies involving NGS via exome or whole-genome sequencing may identify other heme transport and/or degradation genes, which could add to the understanding of the pathophysiology of porphyria and enable the identification of novel targets for porphyria treatment. Recently, a methylation-dependent deep intronic pathogenetic variant was reported to cause EPP [147], suggesting that the non-coding variants detected in clinical genetic screenings should also be evaluated, particularly in the case of symptomatic patients.

However, currently, NGS analysis is being used predominantly for research rather than for diagnosis. Only one study introducing NGS into the diagnosis of porphyria is reported, which involves the validation of a panel, including the entire coding sequences and the exon-intron junctions of the *ALAS1, HMBS, CPOX,* and *PPOX* genes [148]. However, a 100% coverage was reported only for the *HMBS* gene, with the loss of coverage for certain exons of the *CPOX* and *PPOX* genes and a 97% accordance between NGS and Sanger sequencing due to a low percentage of reads (average read-depth) from the insertion/duplication mutant allele. Although the efficiency may be improved by using other panel designs involving a capture rather than the amplicon approach, the cost-effectiveness of an NGS approach for the diagnosis of porphyria in a routine diagnostic laboratory requires further assessment. Moreover, the absence of a public database containing verified pathogenic variants causing porphyria further reinforces that only disease-specific experts would be able to decipher the results of an NGS analysis.

## 6. European Specialist Porphyria Laboratories

The European Porphyria Initiative (EPI), a collaborative network of porphyria centers formed in 2001, evolved in 2007 into the European Porphyria Network (EPNET), where participating centers are required to adhere to agreed quality criteria in order to be classified as specialists in porphyria. The EPNET has also established external quality assessment schemes (EQAS) for laboratory tests starting in 2008. These schemes cover diagnostic strategies, analytical laboratory performances, clinical interpretation, and reporting of results contributing to the improvement of the services offered for the diagnosis of porphyria. EQAS participation is essential for providing quality laboratory diagnostic services and is required for the laboratory accreditation of specialist porphyria centers [149,150].

## 7. Conclusions

Several laboratory tests enabling a precise diagnosis of porphyria are currently available. Simple and inexpensive qualitative screening tests generate valuable information in emergencies, such as an acute attack requiring hospitalization in the cases of ADP, AIP, VP, and HCP or neonatal hemolytic anemia in the cases of CEP and HEP. These tests are useful in all situations in which a rapid diagnosis of porphyria is essential to enable subsequent specific treatments to be commenced as soon as possible and, thus, prevent complications. The quantitative determination of ALA and PBG is important for confirming a suspected attack of acute porphyria, while the other quantitative confirmatory tests play a central role in the precise evaluation of symptomatic patients suspected with any of the different forms of porphyria. Asymptomatic mutation carriers are rarely detected in the measurement of urinary, fecal, and erythrocytes porphyrins, which often demonstrate a high variability and could be just slightly elevated or even within normal limits in the phases between acute attacks. However, the identification of pre-symptomatic carriers is crucial to decrease the risk of overt disease, prevent the long-term hepatic complications of acute porphyrias, and offer genetic counseling for the more severe forms of porphyria. The measurement of enzymatic activities, when possible, allows this issue to be overcome. However, these measurements are somewhat imprecise, as there is a certain overlap among the values measured in patients, the values measured in clinically unaffected gene carriers, and those measured in normal control individuals, thereby rendering the measurement results inconclusive in several instances. In the era of molecular diagnosis, for family screenings, DNA genetic testing remains the preferred method. The routine application of Sanger sequencing and MLPA of the genes that are defective and causing the porphyria is also crucial for the clinicians to obtain the most precise confirmation of a presumptive diagnosis in the index patient of a family. The advent of next-generation sequencing techniques has also enabled molecular biologists to develop a greater understanding of the genes associated with porphyria phenotypes and their functions. However, the possibility of identifying the variants of uncertain significance and the existence of variants missed in the NGS analysis must also be considered. In conclusion, only complementary diagnostic strategies described in the present report are expected to provide the most precise diagnosis of porphyria (Figure 5).

## Figures and Tables

**Figure 1 diagnostics-11-01343-f001:**
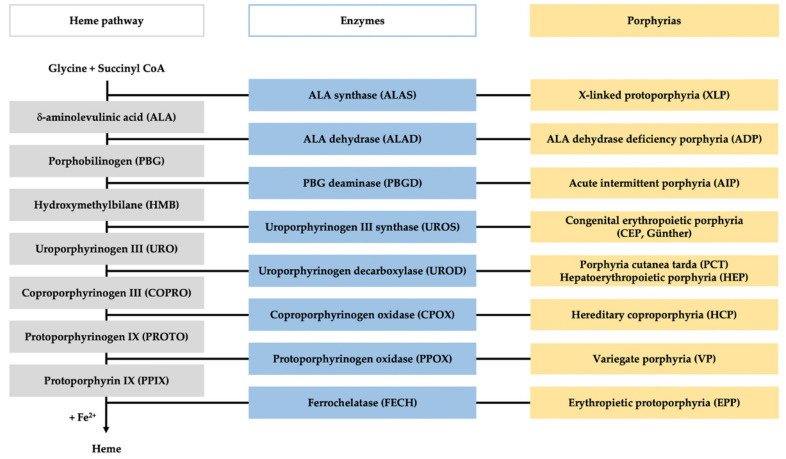
Heme pathway and porphyria. The figure represents the scheme of the heme biosynthesis (in gray), with the involved enzyme (in blue), and the specific porphyria related to each enzyme deficiency (in yellow).

**Figure 2 diagnostics-11-01343-f002:**
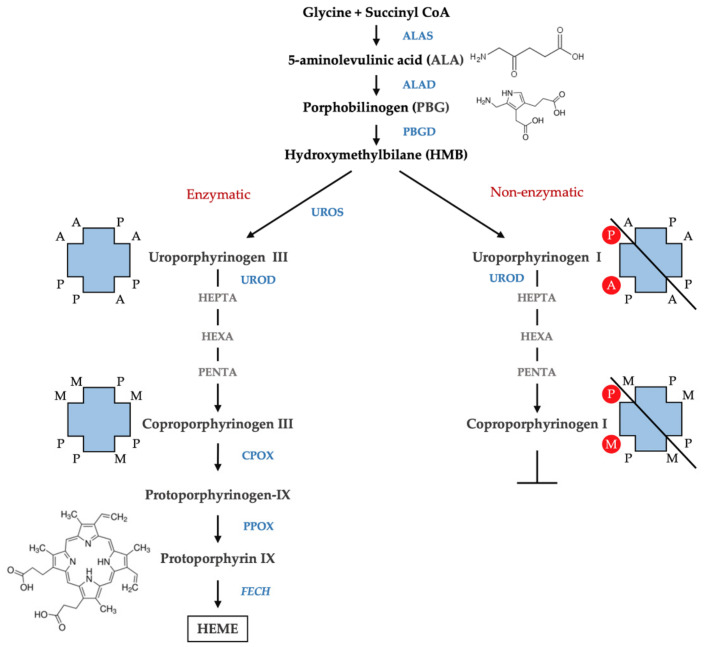
Isomers of porphyrins. The figure represents the scheme of the heme biosynthesis with enzymatic (**left**) and non-enzymatic conversion of HMB (**right**). The arrangement of the substituents of the four pyrroles of the porphyrin ring is shown. A, acetate (-CH2COOH); P, propionate (-CH2CH2COOH); M, methyl (-CH3) and vinyl (CH=CH2).

**Figure 3 diagnostics-11-01343-f003:**
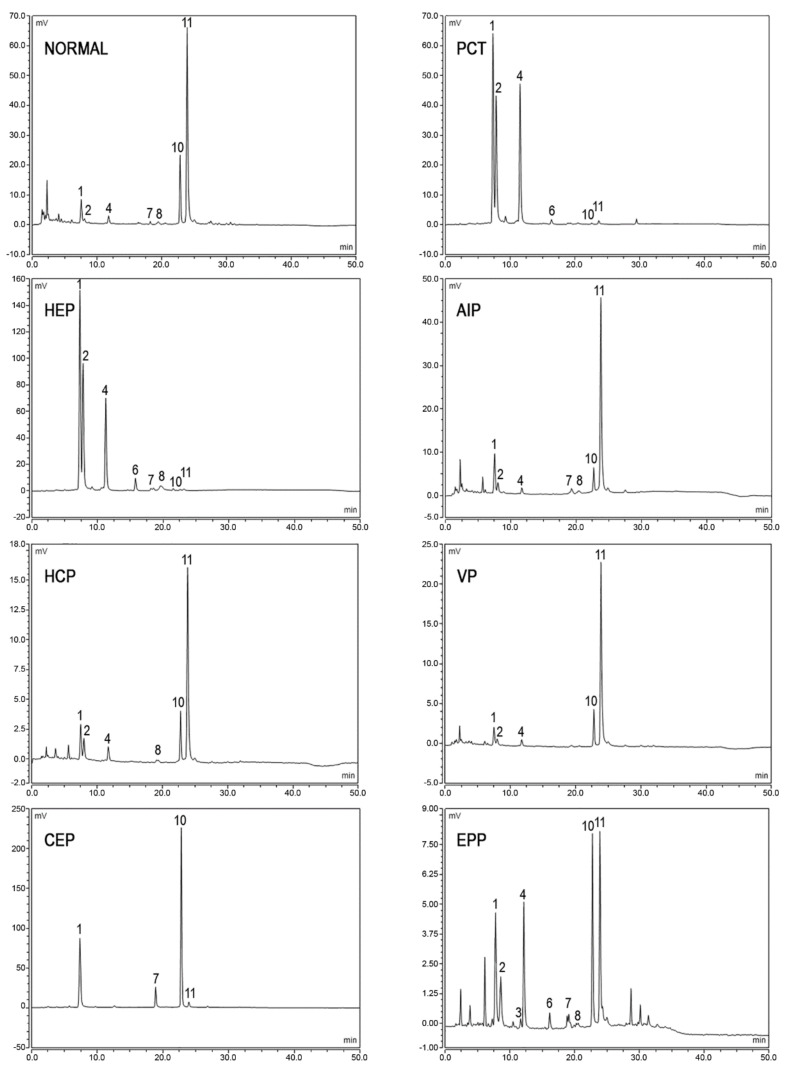
Chromatographic profiles of urine porphyrins. The figure shows the patterns of excretion of healthy individuals (NORMAL) and patients (PCT, HEP, AIP, HCP, VP, CEP, EPP). Fluorescence peaks are identified as uroporphyrin I (1), uroporphyrin III (2), heptacarboxylic acid porphyrin I (3), heptacarboxylic acid porphyrin III (4), hexacarboxylic acid porphyrin III (6), pentacarboxylic acid porphyrin I (7), pentacarboxylic acid porphyrin III (8), coproporphyrin I (10), coproporphyrin III (11).

**Figure 4 diagnostics-11-01343-f004:**
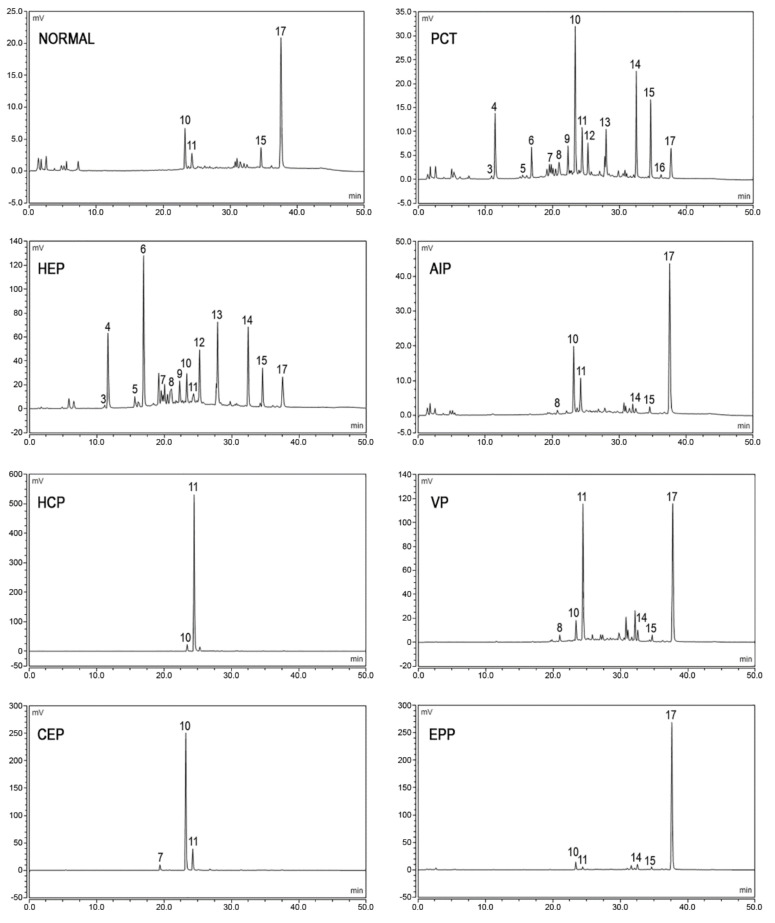
Chromatographic profiles of fecal porphyrins. The figure shows the patterns of excretion of healthy individuals (NORMAL) and patients (PCT, HEP, AIP, HCP, VP, CEP, EPP). Fluorescence peaks are identified as heptacarboxylic acid porphyrin I (3), heptacarboxylic acid porphyrin III (4), hexacarboxylic acid porphyrin I (5), hexacarboxylic acid porphyrin III (6), pentacarboxylic acid porphyrin I (7), pentacarboxylic acid porphyrin III (8), hydroxyisocoproporphyrin (9), coproporphyrin I (10), coproporphyrin III (11), deethylisocoproporphyrin (12), isocoproporphyrin and dehydroisocoproporphyrin (13), deuteroporphyrin (14), pemptoporphyrin (15), mesoporphyrin (16), protoporphyrin (17).

**Figure 5 diagnostics-11-01343-f005:**
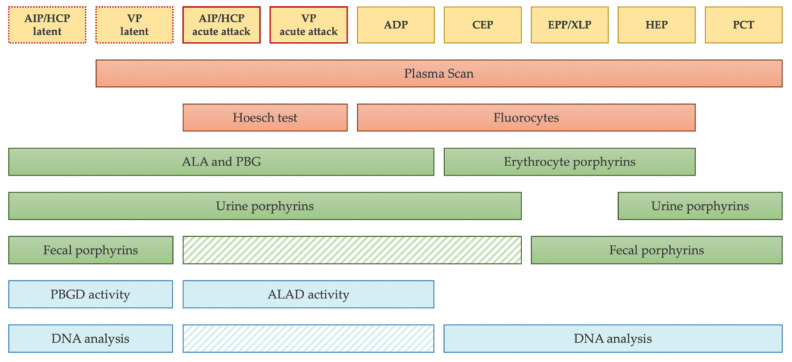
Laboratory diagnostic strategy. The scheme summarizes the appropriate diagnostic testing to prescribe for each form of porphyria. The striped boxes represent the tests that are not mandatory to prescribe before starting a therapy. In particular, fecal porphyrins and DNA analysis could be avoided during acute attack if other tests are positive and they can be postponed to the phases of latency. Of note, in the sporadic form of PCT, no mutation is expected to be found in *UROD* gene.

**Table 1 diagnostics-11-01343-t001:** Maximum fluorimetric emission peak in different porphyrias.

Porphyrias	ADP/AIP/HCP	VP	PCT/HEP/CEP	EPP/XLP
Plasma peak (nm)	618–622	626–628	618–620	632–636

**Table 2 diagnostics-11-01343-t002:** Levels of erythrocyte porphyrins in blood samples of normal and pathological subjects expressed as total concentrations (µg/g Hb) and relative percentage of ZPP and PP-IX.

Porphyrins	Unit	Normal Subjects	Porphyrias	
			EPP	XLP
Total	µg/g Hb	<3	>3	
ZPP	%	85–100%	10–15%	20–40%
PP-IX		<15%	85–90%	60–80%

**Table 3 diagnostics-11-01343-t003:** List of genes causative of porphyria. AD, autosomal dominant; AR, autosomal recessive; XL, X-linked inheritance; ERY erythroid-specific; UBI ubiquitous.

Porphyria	OMIM	Inheritance	Gene	Chr.	kb	RefSeq	Exons	Isoforms
XLP	300752	XL	ALAS2	Xp11.21	22	NM_000032.4	11	Ery
ADP	612740	AR	ALAD	9q32	15	NM_000031.5	13	Ubi/Ery
AIP	176000	AD	HMBS	11q23.3	9	NM_000190.3	15	Ubi/Ery
CEP	263700	AR	UROS	10q26.2	38	NM_000375.2	10	Ubi/Ery
CEP/XLTT	314050	XL	GATA1	Xp11.23	8	NM_002049.4	6	Ubi
PCT/HEP	176100	AD/AR	UROD	1p34.1	3	NM_000374.4	10	Ubi
HCP	121300	AD	CPOX	3q11.2	14	NM_000097.5	7	Ubi
VP	176200	AD	PPOX	1q23.3	5	NM_000309.3	13	Ubi
EPP	177000	AR	FECH	18q21.31	42	NM_000140.3	11	Ubi
EPP2	618015	AD	CLPX	15q22.31	37	NM_006660.5	14	Ubi

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
