# Peer review of "Laboratory Diagnosis of Porphyria"

_diagnostics, 2021, doi:10.3390/diagnostics11081343_

Round 1

Reviewer 1 Report

This is a great review carried out by a demonstrably expert group of authors in the field. This work will undoubtedly be useful both for researchers working in the laboratory, as well as for clinicians who face porphyric patients in person. As for the bibliography, it is exhaustive just as accurate.

Author Response

Thank you

Reviewer 2 Report

Page 3, line 91: ..asymptomatic phase (8). This indicates that elevation of porphyrin precursor levels is a prerequisite but may not be sufficient to cause an attack.

Page 6, lines 256 and 257: creatinine instead of creatine

Here it should be mentioned that for decades 24 hour urine sampling was the rule and then a sample of the collected urine was analyzed.

Meanwhile this procedure has been replaced by the spot urine sample procedure.

See for instance:

Stauch Th, Doss M, Petrides PE, Stölzel, U, Vogeser M. Determination of reference ranges and Cut-off values for the creatinine ratios of porphyrin precursors 5-aminolevulinic acid and porphobilinogen for the reliable evaluation of urine spot samples. International Congress of Porphyrins and Porphyrias, Bordeaux, Frankreich, 2017

Page 7, line 264: again, there a HIGH excreters with no attack.

Page 15, line 599: founder instead of funder

Author Response

Dear reviewer,

Thank you very much for considering a revision of our manuscript. The changes have been introduced into the text according to suggestions. Detailed point-by-point answers to the criticisms are given below.

Hoping that now the paper could be suitable for publication

Best regards

Elena Di Pierro

  • Page 3, line 91: ..asymptomatic phase (8). This indicates that elevation of porphyrin precursor levels is a prerequisite but may not be sufficient to cause an attack.

Yes, it is widely accepted that low levels of precursors may be present in some patients called "high excretors". Of course, these values are noticeable increases in the course of an acute attack.

  • Page 6, lines 256 and 257: creatinine instead of creatine

The term has been changed

  • Here it should be mentioned that for decades 24 hour urine sampling was the rule and then a sample of the collected urine was analyzed. Meanwhile this procedure has been replaced by the spot urine sample procedure.

The following sentence has been added “For decades, the collection of urine samples over a period of 24 hours was the rule leading to a harmful delay of the necessary therapeutic measures. Meanwhile this procedure has been replaced by the spot urine sample procedure that is considered to be sufficient to estimate the activity of acute porphyrias and allow a decision on therapeutic intervention”.

  • See for instance:

Stauch Th, Doss M, Petrides PE, Stölzel, U, Vogeser M. Determination of reference ranges and Cut-off values for the creatinine ratios of porphyrin precursors 5-aminolevulinic acid and porphobilinogen for the reliable evaluation of urine spot samples. International Congress Porphyrias, Bordeaux, Frankreich, 2017

The following sentence has been added “As demonstrated by Stauch Th and colleagues in 2017, in occasion of the international congress of porphyrins and porphyria in Bordeaux, the reference ALA and PBG are considered over the normal limits if the concentration values are >5 and >1.5 µmol/mmol creatinine, respectively”.  However the reference it has not been listed because it cannot be recovered. If the values are different from those indicated in the text, they can be modified as suggested by the referee because there is no other reference.

  • Page 7, line 264: again, there a HIGH excreters with no attack.

Yes, it is widely accepted that low levels of precursors may be present in some patients called "high excretors". Of course, these values are noticeable increases in the course of an acute attack.

  • Page 15, line 599: founder instead of funder

The term has been changed
